# Genetic Background and Clinical Phenotype in an Italian Cohort with Inherited Arrhythmia Syndromes and Arrhythmogenic Cardiomyopathy (ACM): A Whole-Exome Sequencing Study

**DOI:** 10.3390/ijms26031200

**Published:** 2025-01-30

**Authors:** Maria d’Apolito, Francesco Santoro, Alessandra Ranaldi, Sara Cannito, Rosa Santacroce, Ilaria Ragnatela, Alessandra Margaglione, Giovanna D’Andrea, Natale Daniele Brunetti, Maurizio Margaglione

**Affiliations:** 1Medical Genetics, Department of Clinical and Experimental Medicine, University of Foggia, 71122 Foggia, Italy; maria.dapolito@unifg.it (M.d.); alessandra.ranaldi@unifg.it (A.R.); sara.cannito@unifg.it (S.C.); rosa.santacroce@unifg.it (R.S.); giovanna.dandrea@unifg.it (G.D.); 2Department of Medical and Surgical Sciences, University of Foggia, 71122 Foggia, Italy; dr.francesco.santoro.it@gmail.com (F.S.); ilaria.ragnatela@unifg.it (I.R.); natale.brunetti@unifg.it (N.D.B.); 3Cardiology Unit, University Polyclinic Hospital of Foggia, 71122 Foggia, Italy; alessandra.margaglione@gmail.com

**Keywords:** inherited arrhythmia syndromes, arrhythmogenic cardiomyopathy, next-generation sequencing, gene variant

## Abstract

Inherited arrhythmia syndromes include several different diseases, as well as Brugada syndrome (BrS), long QT syndrome (LQTS), catecholaminergic polymorphic ventricular tachycardia (CPVT), and short QT syndrome (SQTS). They represent, together with arrhythmogenic right ventricular dysplasia/cardiomyopathy (ARVD/C), an important cause of sudden cardiac death in the young. Most arrhythmia syndromes are inherited in an autosomal dominant manner, and genetic studies are suggested.: to report the spectrum of genetic variations and clinical phenotype in an Italian cohort with confirmed inherited arrhythmia syndromes and arrhythmogenic cardiomyopathy using whole-exome sequencing (WES). Patients with confirmed inherited arrhythmia syndromes and hereditary cardiomyopathy were recruited at the Cardiology Unit, University Polyclinic Hospital of Foggia, Italy and were included in this study. Genomic DNA samples were extracted from peripheral blood and conducted for WES. The variants were annotated using BaseSpace Variant Interpreter Annotation Engine 3.15.0.0 (Illumina). Reported variants were investigated using ClinVar, VarSome Franklin and a literature review. They were categorised agreeing to the criteria of the American College of Medical Genetics and Genomics. Overall, 62 patients were enrolled. Most of them had a clinical diagnosis of BrS (n 48, 77%). The remaining patients included in the present study had diagnosis of confirmed LQT (n 7, 11%), AR-DCM (n 4, 6.5%), ARVD (n 2, 3%), and SQT (n 1, 1.6%). Using the WES technique, 22 variants in 15 genes associated with Brugada syndrome were identified in 21 patients (34%). Among these, the SCN5A gene had the highest number of variants (6 variants, 27%), followed by KCNJ5 and CASQ2 (2 variants). Only one variant was identified in the remaining genes. In 27 patients with a clinical diagnosis of BrS, no gene variant was detected. In patients with confirmed LQT, SQT, 10 variants in 9 genes were identified. Among patients with ARVD and AR-DCM, 6 variants in 5 genes were found. Variants found in our cohort were classified as pathogenic (6), likely pathogenic (3), of uncertain significance (26), and benign (1). Two additional gene variants were classified as risk factors. In this study, 13 novel genetic variations were recognized to be associated with inherited arrhythmogenic cardiomyopathies. Our understanding of inherited arrhythmia syndromes continues to progress. The era of next-generation sequencing has advanced quickly, given new genetic evidence including pathogenicity, background genetic noise, and increased discovery of variants of uncertain significance. Although NGS study has some limits in finding the full genetic data of probands, large-scale gene sequencing can promptly be applied in real clinical practices, especially in inherited and possibly fatal arrhythmia syndromes.

## 1. Introduction

Progress in genetics analysis on hereditary arrhythmogenic diseases associated with sudden cardiac death (SCD) has allowed a considerable increase in genetic data that have been used in diagnostic, risk stratification, and therapeutic approaches [1]. The developments in next-generation sequencing (NGS) have provided admission to systematic genetic screening, allowing a fast and cost-effective method for the genetic screening of a great panel of genes. NGS allows increasing the mutational spectrum of the genetic variants, contributing to clear phenotypes and pathological mechanisms of disease. Although NGS produce large amounts of data, interpretation of pathological significance still has limitations.

Hereditary cardiomyopathies are a group of heart muscle disorders characterized by heterogeneous phenotypes, including inherited arrhythmia syndromes, dilated cardiomyopathy (DCM), hypertrophic cardiomyopathy (HCM), and arrhythmogenic right ventricular cardiomyopathy (ARVC) [2].

Our cohort consists predominantly of patients with arrhythmogenic diseases like Brugada syndrome (BrS). BrS is a rare, hereditary arrhythmogenic syndrome characterized by the occurrence of ST-segment elevation in the right precordial leads (V1 to V3), referred to as electrocardiogram (ECG) type I. BrS is responsible for SCD in the anatomically normal heart and affects mainly men during the third and fourth decades of life [3]. Although BrS was classically considered as a primary electrical syndrome concerning the sodium channel and leading to the typical ECG, it has been suggested that BrS represents a heterogeneous group of conditions with different genetic and clinical phenotypes. At present, the main genetic cause of BrS is represented by variations in the SCN5A gene, which account for up to 25% of all clinically diagnosed BrS patients. An additional 27 genes have been associated with the disease, but they are considered minor genes representing only approximately 5 to 10% of BrS cases [4,5,6]. Approximately 70% of cases remain without genetic diagnosis.

Inherited arrhythmia syndromes include other diseases besides (BrS), such as long QT syndrome (LQTS), catecholaminergic polymorphic ventricular tachycardia (CPVT), and short QT syndrome (SQTS) [7]. They represent, together with arrhythmogenic right ventricular dysplasia/cardiomyopathy (ARVD/C) and AR-DCM, an important cause of sudden cardiac death in the young [8]. Different genes that have been found to be associated with the ARVC phenotype encode desmosomal and non-desmosomal proteins involved in ARVD development. Genetic heterogeneity with a genetic overlap between heart muscle disorders and channelopathies has been suggested [8].

According to current international guidelines, genetic testing is strongly recommended for every patient with inherited cardiomyopathy to detect a causative mutation and then offer pre-symptomatic analysis of families who are at risk of developing the same disorder [9,10]. To assess the role of genetic variants in arrhythmogenic diseases, we used NGS technology to sequence the main genes responsible for arrhythmogenic disorder in a cohort of 62 patients.

## 2. Results

### 2.1. Clinical Characteristics

We evaluated NGS data of 62 patients (Table 1 and Table 2). In terms of clinical diagnosis, 48 were BrS patients, 6 Long QT, 1 Short QT, 1 with ventricular arrhythmias (VAs), 2 with diagnosis of ARVD and 4 with confirmed AR_DCM. Brugada syndrome is the most common disorder in the NGS group. A total of 48 (77%) unrelated BrS patients were included in this study, of which 73% were male and the mean age at diagnosis was 45 ± 15. Reported family history of BrS and a syncope event were 35 and 14, respectively. Fourteen BrS probands underwent a subcutaneous ICD implantation in primary prevention.

Eight subjects included in the study had a diagnosis of confirmed LQT (n 7, 11%), SQT (n 1, 1.6%). The remaining six patients with a diagnosis of ARVD (n 2, 3%) or AR-DCM (n 4, 6.5%) were also included in our cohort.

### 2.2. Genetic Test

Using the WES technique, 22 variants in 15 genes associated with Brugada syndrome were identified in 21 patients (34%), including SCN5A, CACNA1S, CACNB2, CASQ2, DLG1, JUP, KCNE1, KCNH2, KCNJ5, KCNQ1, PKP2, SCN10A, DSP, MYBPC3, and LAMA2. Among these, SCN5A had the highest number of variants (6 variants, 27%); followed by KCNJ5 and CASQ2 (2 variants). The remaining genes had only one variant (Table 3) [11,12,13,14,15,16,17,18,19,20,21,22,23,24,25,26,27,28,29,30,31,32,33,34,35,36,37].

All SCN5A variants were missense variants, except SCN5A c.935-1G>A, that affect splice acceptor sites. Of the 6 variants, according to evidence aggregated from public databases using ACMG 2015 guidelines, 1 variant was classified as likely pathogenic (LP), 3 variants as pathogenic (P), and 2 variants as variants of uncertain significance (VUS). Two novel SCN5A gene variants were reported in this study, including SCN5A p.(His130Arg) and SCN5A c.935-1G>A. Identified SCN5A gene variants were distributed along the Nav1.5 amino acid sequence as follows: VUS SCN5A p.(His130Arg) and p.(Asp356Asn) in hydrophobic segments (SI); p.(Glu1574Lys) in the transmembrane segment SII; and the remaining in intracellular loops (p.(Gly552Arg) and p.(Glu1053Lys).

All minor genes variants were missense variants. Among the 16 minor gene variants, according to the ACMG 2015 classification, most (13) were classified as variants of uncertain significance (VUS); KCNQ1 p.(Arg195Gln) and MYBPC3 p.(Cys1124Arg) were classified as likely pathogenic. KCNE1 p.(Asp85Asn) was previously described as risk factor [22]. Five novel variants of minor genes were reported in this study. All were missense variants of uncertain significance, including JUP variant p.(Glu138Lys), 2 KCNJ5 variants p.(Thr70Asn) and p.(Asn349Asp), SCN10A p.(Gly1118Ala) and LAMA2 p.(Ile103Val).

Among likely pathogenic variants, MYBPC3 p.(Cys1124Arg) was identified in Patient BrS15. The proband was a 52-year-old male, admitted for diagnosis of BrS. The resting ECG showed a sinus rhythm of 60 rpm with drug-induced type I Brugada ECG pattern and grade I, first-degree atrio-ventricular block (BAV). WES analysis in this subject revealed 2 heterozygous variants: MYBPC3 (p.Cys1124Arg) [27] and DSP (p.Ile874Lys), which were classified LP and VUS, respectively. In patients with spontaneous or drug-induced type 1 ECG pattern (BrP), negative for mutations in SCN5A and other putative BrS genes, likely pathogenic/pathogenic variants in MYBPC3 were found [29]. Variant DSP (p.Ile874Lys) was co-inherited with variation in MYBPC3.

In 8 patients with confirmed LQT, SQT, 10 variants in 9 genes were identified, including KCNH2, CACNA1C, SCN5A, TRPM4, TNNI3K, SCN9A, DCHS1, and KCNQ1. Among 10 variants of genes associated with LQT, SQT, and Vas, 5 novel variants were reported. Of variants identified in 6 LQT probands, 4 were missense, 1 nonsense and 1 a splice-acceptor mutation. According to the ACMG 2015 classification, 5 were classified as variants of uncertain significance (VUS); only KCNH2 p.(Arg744Ter) was classified as pathogenic. A young patient with short QT syndrome had 2 heterozygous variations in KCNH2 p.(Ala1116Val) and p.(Lys897Thr), which were classified as VUS and benign, respectively. Carriership of KCNH2 p.(Lys897Thr), a common single-nucleotide polymorphism that has previously been described as a modifier gene/allele, can affect the clinic phenotype, acting as a susceptibility risk factor for arrhythmia [30,31]. A 69-year-old female (LQT7) carried a heterozygous variation in KCNH2 gene, p.(Asp202Asn), which was classified as pathogenic [32].

Two variants of uncertain significance, previously reported in DSP and PKP2 genes, were identified in 2 patients with ARVD [33,34]. ARVD is principally caused by variants in genes encoding for desmosomal proteins. Variants in the plakophilin-2 gene (PKP2) are the most frequent cause associated with the classical ARVD phenotype. Mutations in the Desmoplakin DSP gene are associated with the manifestation of the disease in 16% of patients [33].

Finally, in 4 AR-DCM subjects, one had a heterozygous nonsense mutation in TRIM63 p.(Gln247*) [35] and 3 had missense variants in VCL, PKP2 [36] and PSEN1 [37]. The heterozygous nonsense mutation TRIM63 (Gln247*) has been associated with hypertrophic cardiomyopathy.

VCL p.(Gly812Ala), a novel variation that has never been reported, was found in a 51-year-old female suffering from cardiomyopathy dilated with sustained ventricular tachycardia. The patient reported numerous syncopal episodes, which is why she was subjected to ICD implantation.

## 3. Discussion

Our study enrolled 62 cases with confirmed inherited arrhythmia syndromes and hereditary cardiomyopathy and conducted WES analysis for these cases to identify potential pathogenic gene variants. The majority of patients in our study were BrS patients.

The worldwide prevalence of the Brugada electrocardiogram (ECG) pattern is estimated to be 0.5 cases in every 1000 subjects, but it is endemic in Asian and Southeast Asian countries, especially in Japan, the Philippines, and Thailand, reaching 3.7 cases in 1000 subjects. Ethnic differences in patients with Brugada syndrome and arrhythmic events were reported [38]. There was no difference in arrhythmic event age onset. Asians were predominantly males (98.1% vs. 85.7% in whites). Asians tended to display a more spontaneous type 1 BrS-ECG pattern (71.5% vs. 64.3%). No difference was observed between the two groups regarding history of syncope. The data obtained from clinical characterization of our BrS subset show consistency with those referring to the European population [38]. The percentage of males in Brs patients is 77%, confirming that arrhythmia is more predominant in males as compared to females in our considered cohort.

In this subsetting, 22 variants in 15 genes associated with Brugada syndrome were identified in 21 patients (34%), including SCN5A, CACNA1S, CACNB2, CASQ2, DLG1, JUP, KCNE1, KCNH2, KCNJ5, KCNQ1, PKP2, SCN10A, DSP, MYBPC3, and LAMA2. Among these, SCN5A showed the highest number of variants (six variants, 27%), followed by KCNJ5 and CASQ2 (two variants). The remaining genes had only one variant. SCN5A gene variants are thought to induce BrS through the abnormal impact on the function of the cardiac sodium channel. Four variants (p.(Glu1053Lys), p.(Asp356Asn), p.(Gly552Arg), and p.(Glu1574Lys) were previously suggested to be causative of BrS [11,12,13,14,15,16]. The remaining SCN5A variants (p.(His130Arg) and c.935-1A>G) are novel; both have an extremely low frequency in GnomAD population databases and a high PhyloP100 score, indicating that both variations occur in a highly conserved site.

Additionally, variants in BrS minor genes, i.e., CACNA1S, CACNB2, CASQ2, DLG1, JUP, KCNE1, KCNH2, KCNJ5, KCNQ1, PKP2, SCN10A, DSP, MYBPC3, and LAMA2, may cause BrS. CACNA1S, located in the chromosome 1q32.1, encodes the Cav1.1 protein, pore-forming an alpha1 subunit of the L-type calcium channel DHPR. The CACNA1S gene was associated with ventricular ectopy and a strong family history of both arrhythmia and familial hypokalaemic periodic paralysis (HPP). An association between the CACNA1S and BrS remains controversial. The CACNB2 gene encodes a subunit of a voltage-dependent calcium channel protein that is a member of the voltage-gated calcium channel superfamily. Mutations in this gene are associated with Brugada syndrome [20].

WES analysis in patient BrS15 revealed the coexistence of two heterozygous variants: MYBPC3 (p.Cys1124Arg) and DSP (p.Ile874Lys), classified as LP and VUS, respectively. MYBPC3 is a structural cardiomyopathy gene and represents the gene most frequently involved in hypertrophic cardiomyopathy (HCM) [39]. In patients with a spontaneous or drug-induced type 1 ECG pattern (BrP), without mutations in SCN5A and other putative BrS genes, likely pathogenic/pathogenic variants in MYBPC3 have been previously found [27]. DPS is a major component of desmosomes, increases TGFB1 protein signalling and results in fibrosis. Gene variants in DSP have been associated with ventricular arrhythmias. DSP gene variants were most frequently represented in BrS patients without structural defects and without any known BrS-associated variants [11,12]. It has been hypothesized that the loss of DSP function may cause a delayed depolarization because of its effects on the reduction in sodium current along with its slow conduction velocity. An association between the DSP gene and BrS remains controversial [13].

Our study also identified several pathogenic or VUS gene variants of CACNA1C, SCN5A, TRPM4, KCNH2, TNNI3K, SCN9A, DCHS1, and KCNQ1 that may lead to LQTs. Among the 10 gene variants identified in patients with LQT and SQT, five novel variants were reported.

In a patient sent for genetic analysis with a suspected LQT, WES analysis highlighted a VUS missense variant in the DCHS1 gene. The DCHS1 gene has been associated with familial mitral valve prolapse (MVP) [40]. MVP could occasionally be associated with serious cardiovascular events, such as malignant arrhythmia, heart failure, or even sudden cardiac death (SCD) [41]. Most patients with MVP do not have evidently recognizable resting ECG anomalies. However, some ECG abnormalities associated with MVP and SCD have been recognized since the 1970s [42]. Some patients with MVP who present with ventricular arrhythmias have longer corrected QT intervals [43]. In this patient, an in-depth clinical investigation highlighted mitral valve insufficiency.

In four AR-DCM subjects, three heterozygous missense variants were identified in VCL, PKP2 and PSEN1 genes along with an additional heterozygous nonsense variant in TRIM63 p.(Gln247*). The heterozygous nonsense TRIM63 variant (Gln247*) was previously associated with hypertrophic cardiomyopathy [38]. Recently, it has been suggested that only homozygous and compound heterozygous carriers developed the full clinical phenotype of the disease, while heterozygous individuals demonstrated mild or almost no phenotype [44].

Finally, a young woman with AR-DCM carried a heterozygous missense VCL novel variation, p.(Gly812Ala), not previously reported. Vinculin (VCL) encodes a cytoskeletal protein, which links actin microfilaments to the intercalated disk and membrane in the heart. VCL was known as a susceptible gene for dilated cardiomyopathy (DCM) and hypertrophic cardiomyopathy (HCM). Studies on mouse models hypothesized that variants in VCL may increase the risk for cardiac conduction defects associated with ventricular arrhythmia without obvious structural heart disease and may account for some cases of sudden arrhythmic death syndrome (SUNDS) [45].

The novel VUS variants of genes in our study were classified according to the ACMG classification. These inherited cardiomyopathies are genetically heterogeneous and phenotypic, and a genetic overlap between cardiomyopathies and arrhythmic syndromes has been documented [46]. However, the predictive outcomes of in silico tools showed considerable discrepancies. Consequently, further pedigree information and additional functional evidence are necessary to confirm the pathogenicity of these variants.

## 4. Materials and Methods

### 4.1. Clinical Investigation

A total of 62 unrelated patients with hereditary cardiomyopathy included in this study were recruited at the Cardiology Unit, University Polyclinic Hospital of Foggia, Italy. Medical and cardiovascular investigation was performed on each patient. Written informed consent was acquired for both clinical and genetic analysis. Medical and genetic investigations were carried out in agreement with the Helsinki Declaration. For patients under the age of eighteen, written informed consent was acquired from a legal representative. The manuscript’s data were all correctly anonymized. The following clinical features were acquired: age, gender and family history of arrhythmic events, which included syncope, sudden cardiac arrest (SCA), and/or ventricular tachycardia/ventricular fibrillation (VT/VF), and a history of prior implantation of an implantable cardioverter defibrillator (ICD). The local ethical committee provided its approval of the study (code of protocol: 3261/CE/20).

### 4.2. Genes of Interest

The Appendix A encloses the gene list for the arrhythmic sudden death syndrome that was studied. The gene list was extended through a search of the recent literature and the Human Gene Mutation Database (HGMD). It contains candidate genes for arrhythmias such as Brugada syndrome and other well-known arrhythmias, in addition to arrhythmic cardiomyopathy gene in vivo models (Appendix A).

### 4.3. Whole-Exome Sequencing

The whole-exome sequencing (WES) investigation was as previously described [47]. Briefly, DNA was extracted from whole-blood samples using the automated extraction instrument MagCore^®^ Plus II following the manufacturer’s instructions. The extracted genomic DNA was fragmented into fragments of 150–200 bp in length, followed by library preparation using the established Illumina paired-end protocol using Illumina DNA prep with enrichment (Illumina, San Diego, CA, USA). The exome-enriched libraries were sequenced on the Illumina NextSeq 550 (Illumina, San Diego, CA, USA). Upon completion of whole-exome sequencing (WES), the quality of the raw reads was assessed, and low-quality reads were removed using the Analysis Toolkit (GATK 1.6). The variants were annotated using BaseSpace Variant Interpreter software, Version 2.17.0.60 (Illumina, San Diego, CA, USA), the reads were aligned to the human genome (GRCh38), and variant calling was done [48].

To predict the functional impact of variants, computational tools were used. In silico possible disease-causing missense variants were evaluated using prediction tools such as Mutation Taster, Polyphen2 and SIFT according the latest recommendations (2022) for PP3/BP4 rules. Population data were obtained from the GnomAD database. Evolutionary conservation in the sequence was established using measurement of PhyloP100 scores based on multiple alignments of 99 vertebrate genome sequences to the human genome. Effects on protein or functional data were also evaluated. These data aggregated from public databases were classified as predicted pathogenic/benign; otherwise, the remaining variants were categorized as having uncertain pathogenicity using ACMG Guidelines. VarSome (https://varsome.com/ accessed on 15 November 2024), ClinVar (https://www.ncbi.nlm.nih.gov/clinvar/ accessed on 15 November 2024), and Franklin by genoox (https://franklin.genoox.com/clinical-db/ accessed on 15 November 2024) were used as tools to sum up actual knowledge about the variants. The molecular validation of variants was performed by standard Sanger sequencing on an automated analyser, SeqStudio (Thermo Fisher Scientific, Waltham, MA, USA

## 5. Conclusions

NGS analysis is a potent approach to systematically discover pathogenic gene variants in many human pathologies. In this study, we used WES to identify novel genetic variations associated with inherited arrhythmogenic cardiomyopathies, namely BrS, SCD, LQT, SQT, ARVD, and AR-DCM. Overall, our results confirmed that arrhythmic patients could carry multiple rare gene variants; in addition, our investigation showed new gene phenotype associations. Furthermore, as our study confirms, data obtained using a WES approach could provide a significant impact on the molecular diagnosis of cardiomyopathies, allowing an early identification of patients at risk for arrhythmia development and a better clinical management of these patients.

## Figures and Tables

**Table 1 ijms-26-01200-t001:** Participant BrS characteristics.

	Overall	Brs (%) *	BrS Wild Type	BrS *SCN5A* Carriers, (%) ^§^	Brs Minor Genes Carriers, (%) ^§^
Probands n	62	48 (77%)	27 (56%)	6 (13%)	15 (31%)
Age at diagnosis (years), mean ± SD		45 ± 15	40 ± 14	49 ± 6	47 ± 15
Male	41 (66%)	33 (69%)	17 (63%)	3 (50%)	13 (87%)
Female	21 (34%)	15 (31%)	10 (37%)	3 (50%)	2 (13%)
Family history, n (%)	52 (84%)	35 (73%)	18 (67%)	6 (100%)	11 (73%)
Syncope	21 (34%)	13 (27%)	4 (15%)	6 (100%)	3 (20%)
ICD	23 (29%)	14 (29%)	7 (26%)	3 (50%)	4 (27%)
Type 1 BrS pattern at peripheral leads, n (%)	30(50%)	30 (64%)	19 (70%)	5 (83%)	6 (40%)

Abbreviations: BrS, Brugada syndrome, * (%) vs. Overall; § (%) vs. BrS group.

**Table 2 ijms-26-01200-t002:** Other participant characteristics.

	LQT Variations Carriers	SQT Variations Carriers	ARVD Variations Carriers	AR-DCM Variations Carriers
**Probands n**	7 (11%)	1 (1.6%)	2 (3%)	4 (6.5%)
Age at diagnosis (years ± SD)	50 ± 17	22	36–60	57 ± 14
Male	2 (28%)	1(100%)	1(50%)	1(25%)
Female	5 (71%)	-	1(50%)	3 (75%)
Family history, n (%)	5 (71%)	1 (100%)	2 (100%)	4 (100%)
Syncope	4 (57%)	-	2 (100%)	3 (75%)
ICD	4 (57%)	-	2 (100%)	3 (75%)

Abbreviations: LQT, Long QT; SQT, Short QT; MVP, Mitral valve prolapse; ARVD, Arrhythmogenic right ventricular dysplasia; AR-DCM, Arrhythmogenic dilated cardiomyopathy.

**Table 3 ijms-26-01200-t003:** Gene variants identified.

ID	Gene	Variant Type	Nucleotide	Protein	dbSNP	phyloP100	Frequencies	ACMG	Evidence	Ref.
BrS1	*SCN5A*	missense	c.3157G>A	p.(Glu1053Lys)	rs137854617	7.905	exomes: ƒ = 0.000154 genomes: ƒ = 0.000138	LP	PM2, PP3, PP5	[11,12,13]
BrS2	*SCN5A*	missense	c.1066G>A	p.(Asp356Asn)	rs199473565	7.905	exomes: ƒ = 0.00000342 genomes:ƒ = 0.00000657	P	PM2, PP3, PP5, PS4, PS3, PM5	[14]
BrS3	*SCN5A*	missense	c.1654G>C	p.(Gly552Arg)	rs3918389	6.169	exomes: not found genomes: not found	VUS	PM2, PP3	[15]
BrS4	*SCN5A*	missense	c.4720G>A	p.(Glu1574Lys)	rs199473620	7.86	exomes: ƒ = 0.000000684 genomes: not found	P	PS4, PS3, PP3, PM2, PP5	[16]
BrS5	** *SCN5A* **	**missense**	**c.583A>G**	**p.(His130Arg)**	**N/A**	**8.011**	exomes: ƒ = 0.00000137 genomes: not found	**VUS**	**PM, PP3**	**N**
BrS6	** *SCN5A* **	**Splice** **acceptor**	**c.935-1G>A**	**---**	**N/A**	**7.312**	exomes: not found genomes: not found	**P**	**PM2,** **PP5,** **PVS1**	**N**
BrS7	*CACNA1S*	missense	c.1493G>A	p.(Arg498His)	rs150590855	7.876	exomes: ƒ = 0.000291 genomes: ƒ = 0.000171	VUS	PM2, PP3	[17]
BrS8	*CACNB2*	missense	c.1873C>T	p.(Arg625Cys)	rs1060499847	3.13	exomes: ƒ = 0.0000157 genomes: ƒ = 0.00000659	VUS	PM2	[18]
BrS9	*CASQ2*	missense	c.532T>C	p.(Tyr178His)	rs1648031031	1.919	exomes: ƒ = 0.0000014 genomes: not found	VUS	PM2, PP1, BP4	[19]
BrS10	*CASQ2*	missense	c.1052A>G	p.(Asp351Gly)	rs200899037	5.729	exomes: ƒ = 0.0000198 genomes: ƒ = 0.0000197	VUS	PM2	[20]
BrS11	*DLG1*	missense	c.1556G>A	p.(Arg519His)	rs141544348	7.905	exomes: ƒ = 0.000512 genomes: ƒ = 0.000362	VUS	PM2,PP1	[21]
**BrS12**	** *JUP* **	**missense**	**c.412G>A**	**p.(Glu138Lys)**	**rs150245906**	**7.896**	exomes: ƒ = 0.0000486 genomes: ƒ = 0.000046	**VUS**	**PM2,** **BP6**	**N**
BrS/ SCD13	*KCNE1*	missense	c.253G>A	p.(Asp85Asn)	rs1805128	2.975	exomes: ƒ = 0.009438	Risk factor	BP6	[22]
BrS14	*KCNH2*	missense	c.2654G>A	p.(Arg885His)	rs1479572342	3.073	exomes: ƒ = 0.000000685 genomes: not found	VUS	PM2, PP3	[23]
**BrS15**	** *KCNJ5* **	**missense**	**c.209C>A**	**p.(Thr70Asn)**	**N/A**	**6.036**	exomes: ƒ = 0.000000684 genomes: not found	**VUS**	**PM2**	**N**
**BrS16**	** *KCNJ5* **	**missense**	**c.1045A>G**	**p.(Asn349Asp)**	**rs774486875**	**3.613**	exomes: ƒ = 0.00000205 genomes: not found	**VUS**	**PM2**	**N**
BrS17	*KCNQ1*	missense	c.584G>A	p.(Arg195Gln)	rs138362632	7.636	exomes: ƒ = 0.00011 genomes: ƒ = 0.0000657	LP	PM2, PP3	[24]
BrS18	*PKP2*	missense	c.2083C>T	p.(Arg695Cys)	rs199583774	1.439	exomes: ƒ = 0.000139 genomes: ƒ = 0.000151	VUS	PM2	[25]
**BrS19**	** *SCN10A* **	**missense**	**c.3353G>C**	**p.(Gly1118Ala)**	**rs868030985**	**0.551**	exomes: ƒ = 0.000000685 genomes: not found	**VUS**	**PM2**	**N**
BrS20	*DSP*	missense	c.2622C>T	p.(Ile874Met)	rs751067479	0.093	exomes: ƒ = 0.0000493 genomes: ƒ = 0.0000394	VUS	PM2, BP7, BP6	[26]
BRs20	*MYBPC3*	missense	c.3370T>C	p.(Cys1124Arg)	rs1360819456	7.071	exomes: ƒ = 0.0000048 genomes: ƒ = 0.0000131	LP	PM2, PP3	[27]
**BrS21**	** *LAMA2* **	**missense**	**c.307A>G**	**p.(Ile103Val)**	**rs369978622**	**8.853**	exomes: ƒ = 0.0000165 genomes: ƒ = 0.0000131	VUS	PM2	N
LQT1	*CACNA1C*	missense	c.6167G>A	p.(Arg2056Gln)	rs112414325	2.576	exomes: ƒ = 0.00234 genomes: ƒ = 0.00195	B	PP2, BA1, BS2, BP6	**N**
LQT2	*SCN5A*	missense	c.3806A>G	p.(Asn1269Ser)	rs761274563	5.109	exomes: ƒ = 0.00000684 genomes: not found	VUS	PM2	[28]
LQT3	** *TRPM4* **	**missense**	**c.3178G>A**	**p.(Ala1060Thr)**	**N/A**	**1.37**	exomes: not found genomes: not found	**VUS**	**PM2, ** **BP4**	**N**
LQT4	*KCNH2*	nonsense	c.2230C>T	p.(Arg744*)	rs189014161	1.2	exomes: ƒ = 0.000000684 genomes: not found	P	PP1, PVS1, PS2, PM2	[29]
LQT/ VT5	** *TNNI3K* **	**splice acceptor**	**c.1028-1G>T**	---	**N/A**	**9.008**	exomes: not found genomes: not found	**VUS**	**PM2 PVS1**	**N**
LQT/ VT5	** *SCN9A* **	**missense**	**c.3736G>A**	**p.(Ala1246Thr)**	**N/A**	**7.905**	exomes: not found genomes: not found	**VUS**	**PM2, ** **PP3**	**N**
LQT/ MVP6	*DCHS1*	missense	c.9115G>C	p.(Ala3039Pro)	rs912004965	7.704	exomes: ƒ = 0.0000098 genomes: ƒ = 0.0000131	**VUS**	**PM2**	**N**
SQT1	*KCNH2*	missense	c.3347C>T	p.(Ala1116Val)	rs199473032	0.489	exomes: ƒ = 0.0000171 genomes: ƒ = 0.0000396	VUS	PM2, PP2	[30]
SQT1	*KCNH2*	missense	c.2690A>C	p.(Lys897Thr)	rs1805123	2.77	exomes: ƒ = 0.325 genomes: ƒ = 0.21	B	BA1/BS2, BP6, PP2	[31]
LQT7	*KCNQ1*	missense	c.604G>A	p.(Asp202Asn)	rs199472702	9.196	exomes: ƒ = 0.00000754 genomes: ƒ = 0.00000657	P	PM3, PS3, PM1, PP2, PM2	[32]
ARVD1	*DSP*	missense	c.242G>A	p.(Cys81Tyr)	rs140965835	3.523	exomes: ƒ = 0.000146 genomes: ƒ = 0.000138	VUS	PM2, BP6	[33]
ARVD2	*PKP2*	missense	c.184C>A	p.(Gln62Lys)	rs199601548	1.915	exomes: ƒ = 0.00027 genomes: ƒ = 0.000387	VUS	PM2, PP1	[34]
AR-DCM1	** *VCL* **	**missense**	**c.2435G>C**	**p.(Gly812Ala)**	**N/A**	**3.105**	exomes: not found genomes: not found	**VUS**	**PM2, ** **PP3**	**N**
AR-DCM2	*TRIM63*	nonsense	c.739C>T	p.(Gln247*)	rs148395034	0.851	exomes: ƒ = 0.000502 genomes: ƒ = 0.000551	P	PM3, PS3, PVS1, PM2	[35]
AR-DCM3	*PKP2*	missense	c.1576A>G	p.(Thr526Ala)	rs397516999	1.613	exomes: ƒ = 0.000146 genomes: ƒ = 0.0000526	VUS	PM2, PP3, BP6	[36]
AR-DCM4	*PSEN1*	missense	c.1279A>G	p.(Ile427Val)	rs1398951357	9.23	exomes: ƒ = 0.00000684 genomes: ƒ = 0.00000657	VUS	PM1, PP2, PM2	[37]

Abbreviations: N, novel variation identified in this study; BrS, Brugada syndrome; SCD, sudden cardiac death; LQT, Long QT; SQT, Short QT; MVP, Mitral valve prolapse; ARVD, Arrhythmogenic right ventricular dysplasia; AR-DCM, Arrhythmogenic dilated cardiomyopathy.

## Data Availability

The data presented in this study are available on request from the corresponding author.

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
