# Peer review of "Genetic Background and Clinical Phenotype in an Italian Cohort with Inherited Arrhythmia Syndromes and Arrhythmogenic Cardiomyopathy (ACM): A Whole-Exome Sequencing Study"

_ijms, 2025, doi:10.3390/ijms26031200_

Round 1
Reviewer 1 Report
Comments and Suggestions for Authors
Dear Authors,
in this interesting top and study, some concerns arise:
abstract: methods usually describe the how the population was searched for. nunbers go into the results section.
methods: please describe how the patients were found by whom and where.
results: table 1 you probably mean mean +- sd; another ‘x’ in syncope line.
below tables usually abreviations are explained.
it is not necessary to give numbers for two of two genders (table 1,2).
for ARVD, you should not give a SD when n = 2, give the two numbers instead.
table 3 is hard to read due to line breaks.
discussion: please compare to other populations (european and non-european) also with regard to the area in Italy where the respective patients came from.
conclusion: usage of references should be avoided, use the discussion for that.
In summary interesting topic. the corrections need to be implemented.
Author Response
Comment 1: abstract: methods usually describe the how the population was searched for. numbers go into the results section.
Response 1: as requested, numbers of patients enrolled are now presented in the Results section of the Abstract (line 26).
Comment 2: methods: please describe how the patients were found by whom and where.
Response 2: how patients were recruited and where is now presented in the Methods section (lines 89-90).
Comment 3: results: table 1 you probably mean mean +- sd; another ‘x’ in syncope line.
Response 3: We apologize for the typo. Table 1 has been corrected.
Comment 4: below tables usually abbreviations are explained.
Response 4: As suggested, abbreviations are now explained at the end of each table.
Comment 5: it is not necessary to give numbers for two of two genders (table 1,2).
Response 5: tables have been amended as suggested.
Comment 6: for ARVD, you should not give a SD when n = 2, give the two numbers instead.
Response 6: as suggested, the table has been corrected.
Comment 7: table 3 is hard to read due to line breaks.
Response 7: according to the Reviewer’s suggestion, Table 3 has been modified to make reading easier.
Comment 8: discussion: please compare to other populations (european and non-european) also with regard to the area in Italy where the respective patients came from.
Response 8: According to the Reviewer’s suggestion we compare our data to other population (European and non-European) in the Discussion section (lines 227-337).
Comment 9: conclusion: usage of references should be avoided, use the discussion for that.
Response 9: As the Reviewer suggested, the conclusion was modified and references quoted in discussion section.
Reviewer 2 Report
Comments and Suggestions for Authors
The study by Maria et al focusses on detecting the various genetic variants and clinical phenotype in patients with Inherited arrhythmia syndrome and ACM in Italian cohort using whole exome sequencing. The study looks sound for publication, yet I have the following comments:
# Why didn't you use healthy controls against the confirmed cases of different arrhythmias to undergo WES analysis?
# In the results, you have shown that the percentage of males among these patients are 66% and in particular in Brs patients as 77%. Does that imply that arrhythmia is more predominant in males as compared to females in your considered cohort?
Author Response
Comment 1: Why didn't you use healthy controls against the confirmed cases of different arrhythmias to undergo WES analysis?
Response 1: in our routinary pipeline, we use a series of patients (after having obtained an informed consent) enrolled for different diseases as standard background for the analysis of genetics findings. Thus, gene variants with an elevated allele frequencies in our population (as SNPs) were not considered for further evaluations.
Comment 2: In the results, you have shown that the percentage of males among these patients are 66% and in particular in Brs patients as 77%. Does that imply that arrhythmia is more predominant in males as compared to females in your considered cohort?
Response 2: findings of a higher prevalence of men in the present cohort is now presented in the Discussion section (lines 235-237).